# Gluteal Compartment Syndrome and Rhabdomyolysis after Prolonged Laparoscopic Nephroureterectomy and Treatment Strategies Including Rehabilitation: A Case Report

**DOI:** 10.3390/healthcare10010047

**Published:** 2021-12-28

**Authors:** Jae-Gyeong Jeong, Seock Hwan Choi, Ae-Ryoung Kim, Jong-Moon Hwang

**Affiliations:** 1Department of Rehabilitation Medicine, Kyungpook National University Hospital, Daegu 41944, Korea; cloud90524@naver.com (J.-G.J.); ryoung20@hanmail.net (A.-R.K.); 2Department of Urology, School of Medicine, Kyungpook National University, Daegu 41944, Korea; skhwan@gmail.com; 3Department of Rehabilitation Medicine, School of Medicine, Kyungpook National University, Daegu 41944, Korea

**Keywords:** compartment syndrome, rhabdomyolysis, nephroureterectomy, surgery, rehabilitation

## Abstract

Background: Rhabdomyolysis is a clinical symptom caused by the rapid release of intracellular components such as myoglobin, lactate dehydrogenase, and creatine kinase into the blood circulation. It is commonly caused by muscular injury including compartment syndrome, infection, drugs, etc. Although it rarely occurs during surgery, the incidence may increase if risk factors such as long operation time, improper posture, and condition of being overweight exist. Case Presentation: A 46-year-old male patient complained of pain and weakness in the right hip area and several abnormal findings were observed in the blood sample, reflecting muscle injury and decreased renal function after prolonged urological surgery. He was confirmed as having rhabdomyolysis, which was caused by compartment syndrome of the right gluteal muscle. After the diagnosis, conservative cares were performed in the acute phase and rehabilitation treatments were performed in the chronic phase. After conservative treatment and rehabilitation, blood sample values returned to almost normal ranges and both level of pain and muscle strength were significantly improved. In addition, about 25 days after discharge, he almost recovered to pre-operative condition. Conclusion: Careful attention is required to prevent intraoperative compartment syndrome. It also suggests that not only medical treatment but also early patient-specific rehabilitation is important in patients with rhabdomyolysis after prolonged surgery.

## 1. Introduction

Compartment syndrome is a condition in which blood flow to the muscles and nerves decreases as the pressure in the muscle compartment increases, which may result in perfusion deficit. Compartment syndrome can be accompanied by rhabdomyolysis as a complication [1]. Rhabdomyolysis is a clinical condition defined as injury and breakdown of the muscle that results in the rapid release of intracellular components such as myoglobin, lactate dehydrogenase (LDH), and creatine kinase (CK) into the blood circulation [2,3]. This situation can lead to several problems, including electrolyte imbalance, metabolic acidosis, coagulopathy, and renal failure [4]. 

In this article, we report the case of a patient who underwent prolonged urological surgery in the right flank position, which was complicated by compartment syndrome with rhabdomyolysis and acute kidney injury (AKI), and a recovery process through rehabilitation treatments.

## 2. Case Report

A 46-year-old male patient weighing 76.9 kg and measuring 1.781 m in height (body mass index (BMI), 24.24 kg/m^2^, overweight) was scheduled for laparoscopic nephroureterectomy. He had a history of hypertension (HTN), alopecia universalis, and alopecia areata. The patient had no other specific clinical symptoms. Left hydronephrosis (grade 3–4) was detected on ultrasound performed for health screening. Then, he first visited a local medical center (LMC) and a left middle ureteral mass with urinary obstruction, suggestive of left ureter cancer, was found on abdominal CT. To establish a treatment plan, he was transferred to the urology department of a certified tertiary hospital. Preoperative urinalysis and blood sample measurements, including blood count, serum electrolytes, blood urea nitrogen, serum creatinine, liver enzymes, and an arterial blood gas study, yielded results that were within normal limits. The patient did not take any nephrotoxic medication. 

On arrival at the operating room, initial vital signs were as follows: blood pressure, 151/91 mmHg; heart rate, 77 beats/min; and arterial oxygen saturation, 100% by pulse oximetry. The total anesthetic time was 5 h 55 min and anesthesia was induced with 150 mg of propofol, 40 mg of 2% lidocaine, supplemental remifentanil continuous infusion, and desflurane, with rocuronium bromide as a muscle relaxant. The patient was first placed in the lithotomy position for ureterorenoscopy and then the position was changed to the right down flank position for laparoscopic left nephroureterectomy. Adequate padding with a sponge and gel pad was applied to the dependent parts of the body. 

The patient was hemodynamically stable (heart rate (HR), 60–80 bpm, and systolic arterial blood pressure, 90–120 mmHg), with all ventilation and oxygenation variables preserved within normal limits. The total fluid input was 2550 mL and the total amount of urine over the procedure was not recorded but within the normal range.

At the end of surgery, the residual neuromuscular block was reversed with 200 mg of Sugammadex, the patient was extubated uneventfully, and he was moved to the recovery room where recovery was uneventful. Post-surgical examination revealed the following: aspartate transaminase (AST), 77 U/L; alanine transaminase (ALT), 38 U/L; blood urea nitrogen (BUN), 15.8 mg/dL; serum creatinine, 1.38 mg/dL; and Modification of Diet in Renal Disease Estimated Glomerular Filtration Rate (MDRD eGFR), 55 mL/min/1.73 m^2^.

The next day, the patient complained of severe pain in his right buttock region, which was edematous. This region was the dependent portion during the operation. The following laboratory examination findings were noted: AST, 666 U/L; ALT, 210 U/L; BUN, 23.7 mg/dl; serum creatinine, 1.28 mg/dL; MDRD eGFR, 61 mL/min/1.73 m^2^; creatine kinase (CK), 35,118 U/L; lactate dehydrogenase (LDH), 918 U/L; myoglobin, 5830 ng/mL; calcium, 8.1 mg/dL; and phosphorus, 1.7 mg/dL. Based on these findings, the patient was diagnosed with rhabdomyolysis resulting from compartment syndrome of the dependent region due to prolonged surgery at a fixed position.

Although acute compartment syndrome was strongly suspected clinically, no definite neurologic deficit was observed in the early stages and the symptoms were relatively mild. Close monitoring and re-evaluation were performed at about 30 min intervals. At this time, since clear signs of worsening of clinical symptoms were no longer observed, it was decided to conduct conservative care and follow-up rather than immediately performing invasive fasciotomy. The patient was also informed about the possibility of performing emergency fasciotomy when symptoms worsened and, with consent, conservative care was continued.

Conservative care with massive hydration (0.9% normal saline, 100 cc/h), hepatotonic agents including Ursa (100 mg tid), and Pennel (25 mg/50 mg tid) were administered orally. Adelavin (3 mL) and Liveract infusion (15 g in 1000 mL of ‘Plasma solution A’) were administered through an intravenous line. Follow-up laboratory examinations were performed daily up to 7 days after surgery (Table 1). Urinary output was monitored and was consistently 990–2750 mL each day.

Soft tissue ultrasonography (USG) and pelvic magnetic resonance imaging (MRI) were performed on postoperative day 7. Generalized edematous swelling in the subcutaneous layer of the right buttock was found on USG with a geographic signal change of about 15 cm (longitudinal) in the right gluteus maximus, medius, and minimus muscles; edematous changes in these muscles and in the overlying subcutaneous layer were observed on pelvic MRI (Figure 1). These laboratory examinations and imaging findings were consistent with rhabdomyolysis from compartment syndrome.

Despite conservative treatment, the swelling and buttock pain persisted. Therefore, he received physical therapy and modality (hot pack, infrared irradiation) after consulting the Department of Rehabilitation Medicine. Laboratory findings gradually recovered as the days passed. On postoperative day 14, finally, measurements of the blood samples returned to a nearly normal range (AST, 15 U/L; ALT, 21 U/L; BUN, 8.8 mg/dL; serum creatinine, 1.02 mg/dL; MDRD eGFR, 79 mL/min/1.73 m^2^; CK, 100 U/L; LDH, 281 U/L; and myoglobin, 73 ng/mL). 

Nineteen days after the diagnosis of rhabdomyolysis, he was referred to the Department of Rehabilitation Medicine for rehabilitation of the right buttock pain and for strengthening of the right hip flexor, extensor, abductor, and adductor muscles. On initial physical examination, the manual muscle test (MMT) showed mildly decreased motor ability, grade 4, in the right hip flexion, extension, abduction, and adduction. The Berg Balance Scale (BBS) was 55 and the Modified Barthel Index (MBI) was 98. The limitation of motion (LOM) of the hip joint suspected of originating from the pain was also observed. When passively moving the hip joint, the patient complained of pain in the gluteal area with a numerical rating scale (NRS) of 5. We interpreted that the patient’s muscle strength and functional level may have been underestimated due to the pain and deconditioning. Rehabilitation treatment was performed for the purpose of improving muscle strength and balance, recovering range of motion (ROM), and for reducing pain based on the patient’s symptoms. Then, we treated him twice a day using an ergometer, sliding rehabilitation machine, and active range of motion (AROM) exercises. Stretching exercises were recommended for strengthening and recovering ROM, and infrared irradiation (IR) and transcutaneous electrical nerve stimulation (TENS) were performed for relieving pain and suspected posterior femoral cutaneous neuropathy. All physical therapy was performed under the supervision of a physical therapist and immediate feedback was provided to improve gait endurance and balance. In addition, the intensity of the exercise was gradually increased according to the performance of the patient. The patient was discharged after undergoing rehabilitation for 17 days. Right hip muscle strength (grade 4 to 5), gait balance, ROM (hip flexion 90′ to 110′; hip extension 5′ to 10′; and hip abduction 20′ to 40′), and pain (NRS 5 to 2) were significantly improved when he was discharged.

## 3. Discussion

Although uncommon, there are cases of rhabdomyolysis originating from compartment syndrome after surgical procedures performed for a long time in an improper position. It is most commonly caused by muscular trauma, including compartment syndrome, and could be caused by muscle enzyme deficiencies, infection, drugs, toxins, endocrinopathies, and electrolyte imbalance [5,6]. Surgical positions that can cause rhabdomyolysis include the lateral decubitus and lithotomy, as well as sitting, knee-to-chest, and prone positions [7]. In urologic surgery, there are few cases of rhabdomyolysis after surgery [8,9]. The flank position is used in nephroureterectomy because it offers better accessibility. However, as the dependent portion is pressed for a long time, circulation to the area decreases and the possibility of compartment syndrome is increased. Rehman et al. [10] recommended bending the table as little as possible during the surgery and bringing the table to a neutral position after retrieving the kidney. Deane et al. [11] also recommended that the degree of flexion could be reduced because there is evidence suggesting that the interface pressure is three times that of the half flexion table when using the full flexion operating table. Some authors suggest that providing adequate cushioning of the gluteal area during surgery may help prevent compartment syndrome [12,13].

It is also important to check whether the patient has risk factors that might induce rhabdomyolysis before surgery. Known risk factors for rhabdomyolysis after surgery include being overweight, long operative time, the lateral decubitus position, extracellular volume depletion, male sex, diabetes, and chronic kidney disease [14,15]. In this case, we believe that direct and prolonged compression of the gluteal muscles against the operating table, accompanied by the patient being overweight and of the male sex, could be responsible for rhabdomyolysis in him. 

There are several criteria for diagnosing rhabdomyolysis but the gold standard for laboratory diagnosis is to check plasma creatine kinase (CK) levels [16]. Although the reference value is not established, in general, a concentration five times the upper normal limit is used as a reference [17]. If a diagnosis of rhabdomyolysis is confirmed, intensive treatment is required, including early massive hydration to prevent end organ damage [18]. Additionally, nephrotoxic agents should be avoided [19].

Gluteal compartment syndrome is an uncommon condition characterized by pain and paresthesia [20,21]. Painful hip movement and tense swelling of the buttock, accompanied by severe hip pain at rest, are findings highly suggestive of gluteal compartment syndrome. This syndrome can also lead to sciatic neuropathy and the loss of peripheral pulses [22]. Direct measurement of compartment pressure aids in the diagnosis of compartment syndrome. Conditions in which pressures in the compartments exceed 30 mmHg require fasciotomy [23]. 

Adib et al. [24] suggested that although fasciotomy is the traditional treatment for acute compartment syndrome, there is no difference in the rate of permanent neurological deficit between surgical and medical treatment in patients with gluteal compartment syndrome without initial neurologic deficit. Additionally, fasciotomy did not provide a distinct advantage in mortality. In addition, although fasciotomy can reduce intracompartmental pressure, some authors are concerned that it simultaneously increases the possibility of developing long-term complications such as bleeding, infection, and scarring [25,26].

In this case, we did not measure intracompartmental pressure during the diagnostic procedure, which is considered a limitation. If accurate pressure values were presented, a clear diagnosis of compartment syndrome could have been made. Although intracompartmental pressure was not measured, considering the clinical features and MRI images, it was consistent with gluteal compartment syndrome, causing rhabdomyolysis.

The symptoms were not severe enough to warrant fasciotomy, thus conservative care was continued. The patient complained of a tingling sensation and paresthesia in the posterior thigh and buttocks. Although posterior femoral cutaneous neuropathy was suspected, the patient refused to undergo a nerve conduction study and electromyography, thus this was not confirmed.

After the acute phase of rhabdomyolysis, the patient was referred to the Department of Rehabilitation Medicine. Physical therapy for strengthening and modality, including infrared ray irradiation (IR) and transcutaneous electrical nerve stimulation (TENS) for pain treatment, were achieved. This is not a common case but several articles have reported the use of rehabilitation treatment in the recovery process after rhabdomyolysis [27,28,29].

Randall et al. [30] introduced a rehabilitation program after rhabdomyolysis. The initial rehabilitation program should be established in a direction that can improve the range of motion while gradually increasing the resistance of the movement. If there is no pain or discomfort for 24 h after exercise, isotonic exercise or more intensified strength exercise can be performed. O’Connor et al. [31] recommended a gradual increase in physical activity with sufficient sleep time and follow-up of CK level and urinalysis.

Schubert et al. [32] presented rehabilitation guidelines after treatment for chronic exertional compartment syndrome. After the surgical procedure, it was divided into four phases in consideration of time and the degree of improvement in the patient’s condition. Pay attention to the occurrence of swelling, scar formation, and pain in the lesion, and perform rehabilitation treatment for the purpose of improving range of motion, strengthening, balance, and cardiopulmonary function. As the phase progressed, the intensity gradually increased. Ultimately, the goal was to return to daily life.

In this case, the patient received a total of 17 days of physical therapy that included hip and thigh muscle strengthening using an ergometer, sliding rehabilitation machine, and active range of motion (AROM) exercises, and a recovering of ROM through stretching exercises and balance training. The patient was able to perform ergometer resistance 3 and sliding rehabilitation machine angle 12′ at the beginning of rehabilitation, and ergometer resistance 6 and sliding rehabilitation machine angle 24′ when he was discharged. Hip muscle strength also improved from grade 4 to 5. In addition, pain was improved from a numerical rating scale (NRS) of 5 to 2 while, simultaneously, the ROM of the hip joint was restored to near normal conditions. BBS and MBI also improved from 55 to 56 and 98 to 100, respectively. The patient was seen at the outpatient clinic 25 days after being discharged and subjectively answered that he was in almost the same condition as before the operation.

Although the patient’s musculoskeletal symptoms were not severe in this case, we believe that early patient-specific rehabilitation treatment could help the patient recover to a normal condition more quickly. Usually, only medical treatments are focused on managing for rhabdomyolysis patients, but it is necessary to pay attention to rehabilitation treatments.

## 4. Conclusions

In conclusion, surgeons should check, prior to the surgery, for risk factors that may induce rhabdomyolysis. During surgery, it is necessary to select an appropriate posture which applies the minimum amount of pressure to the dependent portion and it is also necessary to place the gluteal padding appropriately. If compartment syndrome and rhabdomyolysis occur after surgery, aggressive treatment, including massive hydration, should be performed to prevent kidney injury. Finally, patient-specific early rehabilitation treatment should be administered at an appropriate level to recover the range of motion and deconditioning state, and to reduce pain after the acute phase of rhabdomyolysis.

## Figures and Tables

**Figure 1 healthcare-10-00047-f001:**
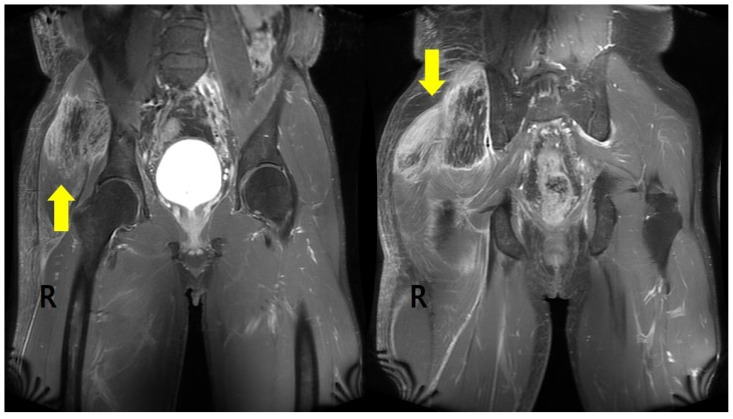
Pelvic MRI axial view shows extensive muscle edema with necrosis involving the right gluteal muscles (arrow) compared with the left side.

**Table 1 healthcare-10-00047-t001:** Sequential laboratory values after surgery.

Variables (Normal Values)	POD ^1^ 1	POD 2	POD 3	POD 4	POD 5	POD 6	POD 7
AST ^2^ (<40)	666	626	615	511	298	193	124
ALT ^3^ (<41)	210	214	223	244	200	167	137
BUN ^4^ (6.0–20.0)	23.7	23.5	17.6	13.3	9.6	9.9	9.4
Creatinine (0.7–1.20)	1.28	1.17	1.07	1.02	1.01	1.06	1.07
MDRD eGFR ^5^ (60–)	61	67	74	79	80	75	74
CK ^6^ (39–308)	35,118	-	26,721	18,997	10,930	7166	4553
LDH ^7^ (<250)	918	-	1045	1131	957	900	788
Myoglobin (14–106)	5830	-	623	296	148	97	119

^1^ Postoperative day; ^2^ aspartate transaminase; ^3^ alanine transaminase; ^4^ blood urea nitrogen; ^5^ Modification of Diet in Renal Disease Estimated Glomerular Filtration Rate; ^6^ creatine kinase; and ^7^ lactate dehydrogenase.

## Data Availability

The data presented in this study are available on request from the corresponding author. The data are not publicly available due to patient’s privacy.

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
