# Peer review of "Gluteal Compartment Syndrome and Rhabdomyolysis after Prolonged Laparoscopic Nephroureterectomy and Treatment Strategies Including Rehabilitation: A Case Report"

_healthcare, 2021, doi:10.3390/healthcare10010047_

Round 1

Reviewer 1 Report

The manuscript produced by Jeong et al., is an interesting and useful addition to the clinical practice of rhabdomyolysis. 

The strengths of the work include:

  1. Rhabdomyolysis is a pathological condition, diverse in etiology and severity of clinical manifestations, in the development of which numerous factors can participate. As a rule, the prognosis for rhabdomyolysis is favorable with a timely diagnosis, however, the severe development of the disease can lead to acute renal failure and even death of the patient. Therefore, doctors of different specialties should take into account the likelihood of its development in case of an unexplained deterioration in the patient's condition, especially if there are factors contributing to this.
  2. The authors present a case of rhabdomyolysis after surgery that is rarely described in the literature. It is important that the manuscript describes in great detail the patient's condition before and after the operation. This allows, along with the main factor (right gluteal muscle separation syndrome), to highlight additional factors (excess weight and sex) in the development of rhabdomyolysis. The results of laboratory tests are also given in detail, the key of which are the level of myoglobin and creatine kinase, on the basis of which the diagnosis was made.
  3. Along with the widely used conservative methods of treatment for mild cases of rhabdomyolysis, the authors have successfully applied physical therapy, which will certainly help the patient recover to a normal condition more quickly.
  4. Clear recommendations in the “Conclusions”.

My suggestions for improvement would be minor textual modifications as follows: -lines 80-81: indicate in the text what CK and LDH mean - line 95 indicate in the text what MRI means -line 112 “On October 8th” must be replaced with the number of days after surgery or diagnosis.

Reviewer 2 Report

  1. General comments

The authors present a case of rehabilitation for a patient with postoperative gluteal compartment syndrome and rhabdomyolysis. I think that the results are not well discussed and conclusion are not meaningful in manuscript.

  1. Specific comments
    1. Major
      1. The present report lacks novelty. It is unclear what the authors want to report in this paper.
      2. The essential treatment for compartment syndrome is a fasciotomy. I think it is necessary to show why it was not required in this case. I am doubtful that this was a case of compartment syndrome.
      3. In the abstract section, you say “it rarely occurs during surgery even in the improper position.” (Lines 15-16), but in the discussion section, you say, “Known risk factors for rhabdomyolysis after surgery include being overweight, long operative time, lateral decubitus position…,” (Lines 139-140). I think these are contradictory.
      4. I think you should discuss a little more about what the rehabilitation interventions focused on and how they might have affected the patient’s outcome.
      5. There are many things in the conclusion section that have not been discussed before.
  1. Minor
      1. Date and place entries in the case presentation section may lead to patient identification and should be avoided.

Reviewer 3 Report

Rhabdomyolysis as a result of improper position is quite rare during surgical procedures. However, if it develops, it can lead to unnecessary complications increasing the hospital stay and the recovery time. The manuscript draws attention to the importance of preoperative control of known risk factors, selecting an appropriate posture, choosing aggressive treatment and it also emphasizes the importance of rehabilitation. In doing so, the case report fulfills its role by providing an important message to clinicians.

A limitation of the case report is that the supposed compartment syndrome was not proven by pressure measurement.

Round 2

Reviewer 1 Report

No comments.

Reviewer 2 Report

It is still questionable whether this case was compartment syndrome or not, but the author has answered all the criticisms raised by this reviewer.

This manuscript is a resubmission of an earlier submission. The following is a list of the peer review reports and author responses from that submission.